# Occupational Violence Experienced by Care Workers in the Australian Home Care Sector When Assisting People with Dementia

**DOI:** 10.3390/ijerph20010438

**Published:** 2022-12-27

**Authors:** Atticus Maddox, Lynette Mackenzie

**Affiliations:** Faculty of Medicine and Health, School of Health Sciences, University of Sydney, Camperdown, NSW 2006, Australia

**Keywords:** nurses, home care services, BPSD, job stress, sexual harassment, physical violence

## Abstract

Background: People with advancing dementia may be dependent on community services from home care workers and nurses to be supported at home. However, these care workers face difficulty undertaking their roles due to challenging behaviours or occupational violence. This study aimed to explore the challenges faced by home care workers and nurses working with people diagnosed with dementia in the community, to identify job demands contributing to their vulnerability to occupational violence, and to determine ways to help manage occupational violence. Methods: A qualitative descriptive study was conducted by interviewing 10 homecare workers and six registered nurses from agencies in South Australia and New South Wales, Australia. Interviews were audiotaped, transcribed and inductive thematic data analysis was conducted. Results: The following themes were identified: (i) sources of threats; (ii) categories of violent, threatening or challenging behaviour; (iii) aggravating factors; (iv) early warning signs; (v) education and training; (vi) managing occupational violence, (vii) resources, (viii) outcomes associated with exposure to occupational violence. Conclusion: Serious issues were identified by participants, yet very little is known about occupational violence for these community care workers. Findings can inform what aspects of work design can be improved to moderate the effects of occupational violence exposure or mitigate rates of exposure, to enable long-term services for people with dementia.

## 1. Introduction

Care work in the home occurs in isolated and unregulated settings where there is a high level of interdependence between the home care worker or nurse and the client. For the purposes of this paper, home care workers are referred to as people who provide regular assistance with tasks in the home. When the paper refers to either home care workers or nurses, the term used is care worker.

The client is vulnerable because of their medical condition and functional dependence on the provision of care services on the home. Conversely, the care worker may be vulnerable due to job insecurity, poor job design, exposures to risks and hazards in the home workplace, as well as weak regulatory oversight and protection. The typical Australian home care worker is a female aged 50 years or over [1] with limited transferable skills relevant to the industry [2]. A significant proportion of the paid home care workforce is casually employed [3] and underemployment is higher than other sectors [4]. There is also evidence to suggest that Australian home care workers may lack adequate training, preparation, and support [5] to perform work that is hazardous [6,7]. The vulnerability of home care workers is increased by the invisible nature of their work [8] which occurs behind closed doors and is not afforded equivalent protection to counterparts working in institutional settings [9]. This group of workers also tends to have low rates of unionisation which may limit their collective voice to improve work conditions.

Care workers may be persistently exposed to forms of occupational violence [5]. Work Health and Safety legislation and accompanying regulations set out procedures for dealing with occupational violence as a generic risk, but do not refer to minimum training, licencing provisions, or account for the complex care needs of persons diagnosed with disabling conditions such as dementia. These regulatory gaps are a concern, given that care workers represent a vulnerable employment group. It is within this context that care workers are at risk of developing physical and psychological injuries during their employment. Therefore, it is important to consider how well care workers cope with this exposure over time and in circumstances where work is performed in isolation and resources are limited.

A pilot study involving a sample of homecare workers in South Australia suggested that occupational violence was pervasive [5]. However, accurate statistics regarding the extent or nature of the problem are not readily available. The term violence in the context of a workplace is also ambiguous and subjective [10,11]. Actual exposure to violence may not be reported and often occurs within a private or hidden context. The magnitude of the problem may be further concealed by an individual’s reluctance to report near-misses or threats of violence. Workers’ compensation data are also unlikely to provide accurate evidence of incidences. Self-employed home care workers may not have taken out insurance to cover themselves for sickness or accidents; they may also be fearful or reluctant to report an incident or be uncertain of their rights.

The negative effects of exposure to occupational violence have been well documented [12,13]. Depending on the nature of the incident and the level of threat that is perceived, individuals may experience anxiety, depression, anger, irritability, sleep disturbance, shock, and loss of confidence [14,15]. In more serious cases, individuals may develop a clinically diagnosable condition such as post-traumatic stress disorder [16]. Over time, frontline care workers (including home care workers and nurses) are at significant risk of developing occupational burnout as a chronic health impairment [12]. Conduct which is identified as threatening, challenging, or violent is representative of behaviours which vary in terms of their seriousness and the challenges that they present. Workplace violence may, for example, arise from sources other than the care recipient.

International studies which have investigated occupational violence involving community care workers providing care for persons diagnosed with dementia, have sought to capture information about behavioural and psychological symptoms of dementia (BPSD) and associated implications for care work [17,18]. Many home care services are offered to people with dementia over the long-term and therefore some instances of violence may be related to the progression of the disease. Some behaviours may be related to the individual with dementia being unable to articulate care needs or they are in environments that are over or under stimulating which exacerbate symptoms [19]. For instance, a bathing activity may be stressful or over-stimulating for individuals with dementia. It may be possible to de-escalate such behaviours by modifying the approach to the task, however, many home care workers or community nursing staff may not be fully trained in such techniques and are thus vulnerable to potentially violent behaviours as a result. Typical tasks homecare workers could undertake with clients include personal care such as assistance with bathing, showering, dressing and eating, everyday mobility and household tasks, whilst registered nurses would conduct clinically related tasks such as medication management, checking blood pressure, managing catheters and wound dressings, all tailored to the client’s needs. These are all very personal tasks which may be undertaken several times a week with each client. Hence, the focus is on experiences of occupational violence for health workers in the home who are the population of interest for this paper.

Most studies of occupational violence experienced by this population have been conducted in institutional settings [20,21] rather than community settings, and used standardised scales [22] to measure variables such as exposure to occupational violence, dimensions of occupational burnout, job satisfaction, job control and workload. Research in this field is limited and occupational violence can be underreported in health and support services [23,24,25]. Despite community and home care workers having the second highest rate of workplace injuries in 2015 [26], it is unclear if injuries reported were associated with physical conditions or psychological (stress-related) conditions associated with exposure to violent, threatening, or challenging behaviour.

There is very little published evidence on the exposure of community healthcare workers to violent, threatening, or challenging behaviour in the homes of their clients. In the absence of a qualitative approach in this area, to explore the experiences of health care workers in depth, research questions included the following: What are the experiences of health care workers when providing care in the homes of people with dementia? Do community healthcare workers have negative experiences that make them vulnerable in their workplaces in the homes of their clients? Do healthcare workers have support from their employer if negative experiences occur? Therefore, this study aimed to (i) explore critical issues and key challenges faced by care workers (home care workers and nurses) who worked with people diagnosed with dementia in the community, (ii) identify job demands perceived by care workers contributing to their vulnerability around occupational violence, and (iii) determine ways perceived by care workers to help manage occupational violence.

## 2. Materials and Methods

### 2.1. Study Design

Using a qualitative description approach [27,28], interviews were conducted with key informants to explore the experience of health care workers in providing care for people with dementia living in the community. This approach is appropriate as the circumstances experienced around this type of work are currently poorly understood. Qualitative interviews were undertaken with consenting individual health care workers to gain an in-depth understanding of their experiences.

### 2.2. Recruitment

Participants were recruited via three consenting home and community care agencies across two states (South Australia and New South Wales). As health services are administered via state governments in Australia, two states were selected to gain a broader understanding of issues for community health care workers who were employed in different state jurisdictions. Participant information statements and consent forms were distributed via the care agencies. Electronic copies of consent forms were sent to participants who agreed to undertake an interview via telephone. Participants were screened for the following inclusion criteria: mandatory minimum age of participation (aged 18 years or above); current employment as a home care worker or nurse providing care to persons diagnosed with dementia; and the location of care within NSW or South Australia. Participants who elected to discontinue an interview for any reason were excluded. Demographic data were also obtained for each participant, including gender, date of birth, level of post-secondary education and employment as a home care worker or registered nurse. The Human Research Ethics Committee (HREC) at The University of Sydney approved the study (2018/1004).

### 2.3. Interview Schedule

The interview consisted of demographic questions to distinguish between home care workers and registered nurses, and to provide information about the diversity of the sample. An opening question was used to address the research aims: *Do you experience violent or threatening behaviour from clients in your homecare work?* Probing questions explored descriptions of this behaviour, how often it happens, and if they consider this is a normal part of their job. Sources of risk and perceived threats, as they are seen by the participants, were explored. Further questions asked if participants knew what to do if a client was violent or threatening, if employers provided guidelines for managing threatening behaviour, what would make it easier to manage violent or threatening behaviour, and what assistance or training was provided. Participants were also asked how confident they were in identifying early warning signs that a client may become violent or threatening, how satisfied they were about their relationships with clients, whether violence has affected their confidence in their role as a caregiver, and a final probing question to explore further issues and/or themes which may be relevant to the study. The meaning of workplace violence was not explicitly defined to enable participants to describe situations they encountered in their work that they considered personally threatening, challenging or violent. The interview schedule was pre-tested with three community health workers not involved in the study prior to implementation for this study. See Table 1 for the interview schedule.

### 2.4. Data Collection

One researcher (AM) conducted all the interviews and reviewed each completed interview to compare with previous findings from prior interviews. This allowed ongoing modifications to future questions and prompts to explore further information where necessary. Care workers’ perceptions that behaviour is threatening, challenging or violent were inherently subjective and interpretations of behaviour varied between individuals. It was important that rapport and trust was established with each interviewee to promote accurate responses, and interviews were non-directive. Interviews were conducted either face-to-face or via telephone. The length of interviews ranged from 38 to 56 min, and they were audio-recorded and transcribed verbatim.

### 2.5. Data Analysis

Data were grounded in the language of care workers and the transcripts were then coded line by line to identify convergent and divergent themes. Data were analysed using inductive thematic analysis [29] that involved consensus coding to develop and refine patterns and relationships between themes and categories. Codes and themes were reviewed four times per transcript, until it was clear which issues had emerged from the data.

## 3. Results

In all, 10 homecare workers (8 male and 2 female) and 6 registered nurses (0 male and 6 female) participated in the study. Participants were aged between 44 and 62 years (Mean = 54 years). Themes from the interviews included (i) sources of threats; (ii) categories of violent, threatening or challenging behaviour; (iii) aggravating factors; (iv) early warning signs; (v) education and training; (vi) managing occupational violence, (vii) resources, (viii) outcomes associated with exposure to occupational violence.

### 3.1. Sources of Threat

Registered nurses and home care workers reported being exposed to violent, threatening, and challenging behaviour from care recipients and, less frequently, from members of the care recipients’ families. Two out of ten home care workers described conduct from care recipients as challenging behaviour (wandering, being complacent, manipulative, or non-responsive to stimuli). These types of behaviour were perceived as distinct from or lower in the spectrum of violent or aggressive behaviours:
*I had one who was threatening, but she wasn’t violent. She was just verbally abusive.*(Home Care Worker H)

Two out of the six registered nurses reported they had been exposed to challenging behaviour from care recipients which had not escalated to higher levels of aggression as care recipients had lower-level care needs.

Three home care workers listed spouses of care recipients or other persons within the home as a recurrent source of threat:
*The client that we’re going in to provide the service for has a friend or relative, or someone like that, in the home …They can be the one that is making us feel uncomfortable.*(Home Care Worker B)

However, two registered nurses disclosed they had been threatened by home care workers:
*It was the client’s [homecare]carer that could be quite aggressive at times…making threats and things.*(Registered Nurse A)

Some interviewees reported that structural/architectural factors, such as confined spaces, may arouse fear and trigger exit strategies when assisting care recipients. This type of reaction tended to occur when interviewees were assisting a care recipient that they were either unfamiliar with or who was known to present with violent or challenging behaviour:
*I’m very aware of my surroundings now when I’m dealing with clients that could pose a threat…I would always make sure I know where the exits are.*(Registered Nurse A)

The presence of threatening objects in the home could also arouse fear and trigger exit strategies in some cases:
*I’ve walked into houses where there’s been a big wooden chopping block with an axe in it. I’ve walked in and I’ve seen that, and I’ve just walked out again.*(Home Care Worker C)

Interviewees perceived that their interpretations about the level of occupational violence they were exposed to were linked to individual factors, such as incidences of prior exposure in their work or private lives:
*You might have a care worker that’s been abused verbally or emotionally all their life…. and then they might go into a situation where family members might yell at one another but that’s normal to them so it’s not challenging.*(Registered Nurse C)

### 3.2. Categories of Violent, Threatening or Challenging Behaviour

In a limited number of cases, conduct involved an overlap between threatening and challenging behaviour. Verbal and physical conduct which was sexual in nature was interchangeably referred to as challenging and threatening. Interviewees had their own privately constructed definitions of occupational violence. For everyone, occupational violence was associated with unwanted physical contact involving a higher magnitude of force, such as kicking or punching.

Overall, a higher proportion of home care workers (six out of ten) reported being exposed to physical types of occupational violence from care recipients, compared to registered nurses (two out of six). Likewise, more home care workers (nine out of ten) described their experience with verbal and other types of non-physical occupational violence.

Categories of occupational violence described were physical or unwanted bodily contact:
*He grabbed me by the throat. I just grabbed his hand. I just looked at him…I didn’t want to like pull his hand away…because he would have got more aggro.*(Home Care Worker G)

More advanced stages of dementia were associated with more violent, threatening, or challenging behaviour among care recipients:
*Going to the third level [of dementia] we’re talking about the baby which is tall and strong….When he’s reacting, sometimes very dangerous for the carer.*(Home Care Worker G)

Six out of ten home care workers and one out of six registered nurses indicated some level of acceptance or anticipation of workplace violence as part of their work:
*Someone will pinch you or you get a thumping or something, but that’s something I know about. Unfortunately, dealing with people with dementia, that’s part of the job. You just hope and pray that it’s not often.*(Home Care Worker A)

Interviewees also described other physical threats such as throwing objects, threats to hit or punch, banging using fists on the table, banging the floor using a walking frame, picking up and throwing television sets at caregivers, frustrated and agitated behaviour, getting backed into a corner, guns in the home or jumping on tables.

Interviewees described unwanted contact or threats of contact of a sexual nature. This type of conduct tended to be described less overtly, such as *inappropriate touching* or *sexually explicit, very touchy*. According to one home care worker:
*You get scared if you go in there…when they look you up and down, like sexual.*(Home Care Worker C)

It was apparent from the interviews that underreporting, poor communication and normalisation of this type of threatening behaviour was common:
*He wasn’t violent, but more of a sexual thing...a couple of girls had told me about his behaviour…but they never reported it…when I went to the house, straight away I just ran away, emailed my coordinator and told her. I never went back.*(Home Care Worker A)

Interviewees also described non-physical, verbal aggression. This included making personal threats, using profanities or language which was personally offensive, accusations of theft, complaints about the quality of care, negative personal comments about the care worker, swearing, yelling, screaming, and using intimidating language. Managing verbal and other types of aggression associated with BPSD was further complicated by a care recipient’s fluctuating mood:
*A few times she’s been really, really nasty and I’ll say to her: ‘Look, I’m not here to be abused, we’re coming here to help you and if you’re going to be like that I’m not going to come home anymore.’ Then she apologised…*(Home Care Worker C)

Frequent complaints made by care recipients caused some registered nurses to question their clinical skills and competence:
*Nothing’s ever right and I change this man’s catheter once a month. Despite the fact that I put all his catheter bag and everything back exactly the same way it was, it’s always wrong.*(Registered Nurse E)

Other people in the home (secondary to the care recipient) were also noted to make demands on care workers or engage in conduct which was personally distressing:
*He was telling them to not talk to the client and just keep cleaning. He wanted all the services to be cleaning services. He was following them around, he threatened to put cameras up to do video surveillance so that he could watch what they were doing.*(Registered Nurse D)

Some behaviours involved a sexual element. Examples cited were lewd jokes, sexual connotations, leering, explicit comments about a care worker’s body and requests for sexual favours. According to one home care worker:
*All he wanted was sex and he kept saying all these horrible things. I was supposed to be cleaning and he said, ‘Don’t worry about it. Sit here next to me.’*(Home Care Worker C)

In several cases, fear and anxiety associated with the sexual element of verbal aggression was exacerbated by other contextual factors:
*He was quite rude in the way he spoke to me ..he made a dirty joke, lewd…he insisted that the dressing be done in the bedroom and I did it in the bedroom but I felt very uncomfortable…*(Registered Nurse D)

All interviewees were careful to distinguish challenging behaviour from violence.

### 3.3. Aggravating Factors

There was general consensus among interviewees that BPSD could be aggravated by a range of factors such as ingesting drugs, being exposed to stimuli such as photographs, television bright lights, colours or noise, disrupting the routine and environment of care recipients, providing assistance with ADLs, forcing rather than negotiating with care recipients to participate in pre-established care routines, the care recipient perceiving an elevated or threatening tone of voice:
*If you raise your voice, they get frightened and they think: ‘She’s attacking me.’ So, that’s when they thrash out as well.*(Home Care Worker J)

Not being adequately prepared to attend to care recipients was noted to have several disadvantages for care workers. In many cases interviewees observed that non-intentional disruptions to a care recipient’s routine may aggravate BPSD:
*It’s really important to find out as much as possible. Because otherwise you’re going in there blind. It’s even simple things like making them a cup of tea and putting sugar in it when they don’t have sugar. That can send them off.*(Registered Nurse F)

A frequent theme raised by interviewees was the need to negotiate and work with care recipients, rather than engage in coercive behaviour:
*If I give medication and they don’t want to take the medication…they’ll hit out with a fist or their legs…I’ve had a few black eyes.*(Home Care Worker J)
*On occasion she can just suddenly become angry and aggressive…I’m always very cautious when I deal with her. If she doesn’t want to get into the shower and we don’t push the issue. But that can be difficult when there’s been no showers for a couple of weeks.*(Registered Nurse D)

A care recipient’s prior exposure to violence at an earlier stage in life may increase the probability that they will exhibit BPSD:
*He grew up in a violent environment. That means if he remembers what he lived years ago and has memories of his behaviour half a century ago, that returns to him.*(Home Care Worker D)

It was apparent from some participant’s observations that socialisation, personality and gender may also factor into a care recipient’s propensity toward violent or threatening behaviour as manifestations of BPSD:
*The violence is…a man’s privilege. The women act differently. They are not violent…Maybe sometimes words or complaints about their families…*(Home Care Worker G)

Interviewees believed that the type and stage of dementia was associated with manifestations of BPSD:
*When looking at alcohol-induced dementia, people are very aggressive or their sun-downing is more pronounced. So, you’ve got an acceleration of symptoms.*(Registered Nurse B)

Five out of six registered nurses offered further clinical insight by explaining that underlying secondary medical conditions such as delirium should be acknowledged and understood:
*If they’re in delirium it’s a different story…you’ve got to find out what’s wrong with them. It could be in their blood, it could be a urinary tract infection...it could be pneumonia, it could be an ingrown toenail.*(Registered Nurse A)

Home care workers with limited training yet extensive industry experience (up to nine years) had acquired practical knowledge which was useful in their work:
*They don’t understand direction….if you were to say sit down, calm down, actually it’s encouraging them to be more violent. So, the best course of action is just to walk away, let them be and then approach them again.*(Home Care Worker J)

However, not having the same care worker could sometimes disorient, confuse, or otherwise threaten care recipients:
*If you haven’t got a stable workforce, you’ve got different people going into an environment that they don’t know…which really does accelerate any behaviour that might happen...*(Registered Nurse B)

### 3.4. Early Warning Signs

The range of markers indicating warning signs included changed facial expressions, evidence of agitation, flickering eyes, the presence of tears or crying, changed tone of voice, shaking, twitching or trembling hands and fists, persistent questioning, pacing and engaging in other types of repetitive behaviour, refusing to co-operate with a care worker, and non-specific types of agitated behaviour. Three out of ten home care workers and two out of six registered nurses alluded to cascading signs and symptoms:
*Her voice became very loud and high-pitched. She was shaking, she was getting more and more distressed, it was increasing. Then she started crying.*(Registered Nurse C)
*If he starts twitching his fists and he starts trembling with…anger. Something’s got to give.*(Home Care Worker G)

Confidence in being able to address care recipient needs or de-escalate BPSD was a convergent theme among registered nurses:
*Even though sometimes I’m upset for a couple of hours after or I’m still thinking about it tomorrow morning or whatever, it’s not going to upset me…they need me, so that’s why I’m here.*(Registered Nurse E)

Home care workers were less confident in being able to identify early warning signs associated with BPSD without prior briefing, instruction, or review of a care recipient’s clinical history. This was called *going in blind*.


*A lot of our work involves going in blind…if I can’t speak to a coordinator that has the case history, ..when I get there what I see is what I get. So, you can go in to some really awful situations sometimes, and we just have to deal with it, or not.*
(Home Care Worker B)

Even if home care workers had long-standing relationships with care recipients, it was generally acknowledged by interviewees that the behaviour of some care recipients could be unpredictable:
*She just tore shreds off me…I just stood there, absolutely dumbfounded. I had no idea what happened. No idea at all.*(Home Care Worker H)

Sudden and dramatic turns of violent behaviour could place home care workers at immediate risk of injury and create challenges for effective and responsive risk management strategies:
*You can tell when a person’s about to hit you. You just walk away, retreat, let them have that five minutes, and then go back and use a different tack…*(Home Care Worker J)

A theme among both occupational groups was an acknowledgement that care recipients should not be held accountable for violent or threatening behaviour associated with BPSD. This remained the case regardless of the intensity of violent conduct or level of harm generated. Registered nurses adopted a clinical lens when discussing BPSD during interviews. They tended to regard the onset of BPSD as an inevitability which could be monitored and managed using clinical skills and judgment. Presentation of unexpected and difficult to manage BPSD among care recipients was viewed by registered nurses as a creative and professional challenge and tended to prompt investigation using a range of resources such as colleagues, family members, professional bodies or online sources. Conversely, several home care workers disclosed feelings of self-blame and guilt for either aggravating BPSD or failing to use effective methods to de-escalate:
*Sometimes I cry inside and I wish I had the knowledge of what to do or say.*(Home Care Worker I)

### 3.5. Education and Training

Relevance of prior education and training in being able to manage occupational violence emerged as a theme among care workers, but not registered nurses. Eight out of ten home care workers reported they had completed some level of training which involved dementia care. However, many indicated there were significant gaps in training. Consequently, approximately 50% admitted to feeling unqualified to effectively manage BPSD by themselves. By comparison, four out of the six registered nurses who had completed post-graduate qualifications (including a coursework component in dementia) reported practical benefits such as knowing how to deal with difficult behaviours, knowing how to act early and read signs of increasing agitation.

Lack of clinical education and training among home care workers was observed by registered nurses to be problematic:
*The staff that are going in haven’t got that much education about dementia behind them. What we’re finding is people are staying at home far longer than they were even ten years ago. So, you’ve got all these behaviours that are harder to manage in a home environment.*(Registered Nurse A)

Limited clinical education and training may result in a discrepancy between the demands of care recipients and the capabilities of home care workers.


*They’ve got staff going in that are inexperienced and they’re going into a high level four client, and doing things like not walking beside them when they’re out walking, or not totally seeing them seated before they turn away.*
(Registered Nurse B)

This may jeopardise the health and welfare of both care recipient and worker:
*If you send someone in that doesn’t know how to manage challenging behaviours –they’ve got minimal training behind them. …In a home, it’s only you and the client. So, you’re expecting this person to be able to manage and I think if you’re sending people in unprepared, it’s just fraught with danger.*(Registered Nurse B)

A lack of clear guidelines for managing this type of behaviour was an apparent problem:
*We have had someone with challenging behaviours, as in sexual connotations and touching and things. That was extremely hard to manage in the community.*(Registered Nurse B)

### 3.6. Managing Occupational Violence

Despite care workers reporting feeling confident in their ability to identify early warning signs that a care recipient may become violent, threatening or challenging, they were less confident in taking the next step to manage this type of hazard. There was a lack of consensus about how to manage challenging behaviours which was attributed to a range of factors such as no specific guidelines or codes of practice to direct home care workers on optimal procedures for managing occupational violence (including BPSD). In the absence of clear and practical guidance, workers employed a range of strategies which drew from a combination of intuition, trial and error and prior training. Problematically, these approaches tended to be ad hoc, mechanistic and were designed to be implemented after an incident rather than as a pre-emptive strategy:
*We do have guidelines for what we can and can’t do. Basically, if it happens, incident reports are done, coordinators are informed immediately and if it’s really life-threatening, then we just back out.*(Home Care Worker A)

Home care workers managed BPSD by identifying early warning signs associated with BPSD, creating distance between themselves and the care recipient and attempting to de-escalate verbally (if possible). Only in severe cases did homecare workers indicate they would contact their manager. Less frequently, an incident report will be completed. Many homecare workers also reported a reluctance to inform family members as they often denied the behaviours.

Home care workers could not use chemical or physical restraint. However, in severe cases, three registered nurses described the strategy of locking doors to confine a care recipient’s violent, aggressive or threatening behaviour.


*Some things will work for one client and won’t work with the next…you just try to come up with a solution and if that works, great. If it doesn’t then you try something else….*
(Registered Nurse A)

This approach was reflected by home care workers with less advanced clinical skills, judgment, education or training:
*All we can do is take each person individually, each situation individually, and deal with it to the best of our ability.*(Home Care Worker B)

Unlike any of the home care workers, four out of six registered nurses felt obliged to consult with colleagues, professional bodies or other, when care recipients failed to respond to strategies designed to manage violent or challenging behaviour:
*I get in touch with the Dementia Advisory Service…they have a lot of ideas. I attend their meetings every couple of months.*(Registered Nurse A)

A further point of difference between the two occupational groups was that registered nurses tended to draw upon clinical training and experience when engaging in de-escalation strategies:
*I’ve worked previously in a mental facility where we’ve had a lot of clients with dementia who at times were quite aggressive…it’s best just to de-escalate any situation. There’s no point being assertive. You need to back off.*(Registered Nurse D)

Home care workers tended to describe tailored and ad hoc approaches to managing BPSD based on prior exposure with a particular care recipient:
*You just walk away, retreat, …nine out of ten times, they will change their behaviour. But there’s always that occasion when you’re not successful…you’ve just got to know your person.*(Home Care Worker J)

In some cases, cultural and language barriers may present challenges for managing BPSD, including de-escalation strategies:
*We’re dealing with an older population that have hearing limitations already and when you put someone where English is not their first language and if they get into a situation where they’re unsure, they will always speak faster than they normally would…so that whole scenario becomes more difficult.*(Registered Nurse B)

Several home care workers noted the difficulty of providing patient-centred care within allocated shifts. It was acknowledged by five out of six registered nurses and four out of ten home care workers that rushing or forcing care recipients to comply with caregiving assistance provoked agitated behaviour. While there was consensus that patient-centred care was instrumental in avoiding or mitigating BPSD, time constraints represented as source of stress for nurses:
*If I need to do something clinical and they’re in the lounge room and I want them to go and lie on the bed because I’ve got to do a dressing…that will often accelerate their behaviour… so instead of it being a 30 min visit, it might blow out to an hour visit…*(Registered Nurse B)

A significant aspect of managing occupational violence (including BPSD) involved the regulation of emotions for the carer:
*What we’re taught is not to exacerbate the situation at all, by yelling back…even though that’s what you’d like to do. You’re supposed to speak…calmly and slowly and with a modulated voice.*(Home Care Worker B)

Emotional regulation was described by some care workers as demanding, draining or stressful. Reference was made to processes requiring cognitive effort, including patience, remaining calm, subverting or suppressing negative emotions, deliberately smiling or demonstrating positive displays of emotions which were inauthentic, using a persistent low tone of voice. Not surprisingly, a proportion of home care workers perceived an inequity between job demands, skill, effort and rate of remuneration:
*I think the demands of the job and the skills that you have are professional…your skills are going to be for a good job in the high level. But who’s going to work for this kind of money?*(Home Care Worker G)

Four out of the ten home care workers interviewed described themselves as tough, thick-skinned or battle-hardened and disclosed that they had been victims of domestic violence in the past:
*I’ve had violence in my life. My mum wasn’t always the nicest. Hers was a belt buckle…I ended up having a husband like it. So, yeah, and it sort of changed me. I treat people the way I’d like to be treated.*(Home Care Worker C)

However, it is uncertain whether prior exposure to violence increased or decreased a home care worker’s vulnerability toward occupational violence or influenced their perceived normalisation of this hazard.

### 3.7. Resources

Nurses described an association between lack of resources and reduced quality and consistency of home care:
*It all comes down to the dollar… the clients we’re seeing in community have higher care needs and higher behavioural difficulty, I think it’s going to become harder. I just know we seem to be running through staff quite quickly…the reality is the job is hard. It’s hard.*(Registered Nurse B)

Government Home Care Packages are fixed funded schemes. It could be a source of tension for care workers when they felt that they were unable to meet the expectations of care recipients or their family members:
*I’m finding family dissention is becoming more common…they want more than is actually available and feasible. That is often a big problem. Their expectations are very high…but the reality is that the funding is not there, so you can’t do it… the expectations of people are getting higher.*(Home Care Worker A)

Induction and preparation for working roles were discussed by interviewees. Three registered nurses questioned if new home care workers were afforded a positive or useful learning experience before commencing work independently. In all, 40% of home care workers reported that training and preparation for home care services could be improved by adopting a hands-on approach and extending the induction period:
*When I have a buddy, I make them do it so that it registers, so it sinks in, and they actually know what they have to manually do…*(Registered Nurse B)

None of the interviewees indicated that tools to measure occupational violence risks were used. Care recipients were mandatorily screened for dementia, and the stage of dementia informed the level of care which individuals may be entitled to receive. However, according to one registered nurse, ACAT assessments tended to lack integrated information, contained biased and outdated content, and frequently did not provide an accurate indication of the care recipient’s current status:
*Unless you’re getting good referring information, you never quite know…until you start spending a greater length of time with someone…the ACAT assessment is quite detailed. But it’s often done in a hospital environment or when there’s crisis. So, you’re not getting a clear picture of the family situation…*(Home Care Worker A)

Registered nurses and home care workers expressed differing attitudes towards completing incident reports. Eight out of ten home care workers were reluctant to report incidents unless they were considered serious. Half of the nurses interviewed felt obliged to document significant incidents of BPSD. Two key reasons were cited: (i) to establish a clinical history which could be used as evidence that dementia was progressing, and (ii) use this evidence to apply for higher levels of funding to reflect higher car needs. One nurse indicated that documenting incidents could also produce a cathartic effect:
*Sometimes you feel a bit overwhelmed…just by documenting what happened and reading over it again…sort of makes it easy.*(Registered Nurse A)

A divergent theme was knowledge of protocols for responding to occupational violence. All the registered nurses and a majority of the home care workers advised that their organisation did have procedures in place for dealing with situations which threatened their perceived safety or wellbeing. Registered nurses on the other hand, cited an overriding duty of care to the care recipient as a factor to consider when determining how or when to employ risk mitigation strategies:
*I was halfway through an insertion of a suprapubic catheter so I couldn’t just walk out the door. If it had escalated any more, I would have [left]…he settled and I was able to just make him safe and then I left. In some situations, it may not be achievable. I hope I never get into that situation.*(Registered Nurse D)

Some interviewees described levels of client attachment. Close personal connections between registered nurses and care recipients were acknowledged to occur despite clinical training:
*Some of them I see more often than others and you develop more of a relationship with them, a bit more of a professional friendship…* (Registered Nurse E)

### 3.8. Outcomes Associated with Exposure to Occupational Violence

Six out of ten home care workers and three out of six registered nurses reported a connection between exposure to occupational violence and consequential feelings of stress and anxiety:
*Dealing with it at the time is very emotionally draining. So, unless you can walk away from the situation and either debrief or go for a walk outside…you do become quite emotionally distressed.*(Registered Nurse F)

Registered nurses also reported high levels of stress associated with managing clinical, administrative, and managerial aspects of work with minimal resources:
*I know I’m good at what I’m doing but it’s getting harder. It’s the type of work… you’re trying to juggle a lot more things. I don’t know whether that’s because of the resources, the funding or the fact that people are staying at home longer, with higher care needs*.(Registered Nurse B)

Ultimately, most home care workers felt they were not adequately supported in their roles:
*It is confronting as a caregiver because in a community setting you’re on your own. You don’t have colleagues that you can call to come into the room and back you up. You have to be fairly confident. You have to be fairly firm.*(Registered Nurse D)

To a lesser extent, the health and wellbeing of home care workers was perceived to be a secondary concern behind the welfare and needs of care recipients:
*I just find a lot of focus is on the client…make sure they’re not injured or abused or neglected or anything like that. But not much emphasis is put on the staff and they are really at risk.*(Registered Nurse C)

Three home care workers and two registered nurses expressed a genuine fear of being seriously injured by a care recipient in the course of their usual work duties:
*I finally said, ‘I just can’t go back in there, I’m scared he’s going to throw hot water in my face or something.’ I was really worried he was going to do something like that.*(Registered Nurse A)

When asked whether violence was considered a normal part of work in homecare, participants tended to elicit normative responses, stating that it should not be part of the job:
*It shouldn’t be something that is acceptable. It’s not ok for us as workers to be exposed to that sort of thing.*(Registered Nurse D)

Some home care workers reported an internal conflict between exposure to BPSD and positive attachment to a care recipient:
*I never refuse to work with someone who’s been aggressive towards me. I always go back for more. We’ve got a man who’s 100. I’ve worked with him for many, many years. He’s the most vile, obnoxious, cantankerous old person…the things he says are awful, but he’s a really nice man, which is really weird.*(Home Care Worker B)

There was disagreement in both groups about whether a care recipient’s violent, threatening or challenging behaviour impacted on a homecare worker’s reported attitude and wellbeing. This may be related to perceived resilience:
*Over a period of time he became more and more aggressive and incoherent and threatening. I got to the point where I refused to go back because he had focussed on me, and he made threats against me…I just couldn’t go back. I was scared for my health and wellbeing.*(Registered Nurse C)

It may also be significant that care workers who admitted to experiencing reduced confidence or mental distress following an incident of workplace violence regarded these outcomes as short-term.


*Sometimes it can put your stress levels through the roof.*
(Registered Nurse A)


*When you’re down, I put my music full blast on in the car and I’m ok until next time.*
(Home Care Worker H)

There was a trend among more experienced care workers to know when to step back from providing direct care once early warning signs are identified in care recipients. They acknowledged that there was not an unlimited expectation to provide care. Registered nurses tended use clinical judgment when making decisions about withdrawing from some activities:
*This morning I went to do a psychogeriatric assessment scale for a client…she was very confused, really agitated and she refused to participate…so I decided to not do the assessment… I could notice signs of increasing distress. It wasn’t going to get any better.*(Registered Nurse C)

There were differences in opinions about support from management. Whereas six out of ten home care workers were reportedly satisfied with the level of support they received from management, the nurses agreed on their lack of organisational support:
*He was really vindictive, and he said some terrible things about me…I had a mediocre amount of support from my managers. They initially took it seriously because I had a third party that verified my concerns. But after a while they decided to acquiesce and put services back in the home even though the care workers said that they were scared of this fellow…I feel like I wasn’t taken seriously.*(Registered Nurse C)

Some home care workers and registered nurses described an intention to leave their role:
*I suppose it’s a job that you don’t want to do forever…a few more years maybe and then that’s it. Because I think you just want the stress out of your life, that’s what it comes down to at the end of the day.*(Home Care Worker A)

Of those who indicated an intention to leave their role, work–life conflict featured as a theme:
*Some days you’re still talking about it at 10 o’clock at night with your glass of wine in your hand…I have times when I’m awake at 3 o’clock in the morning for whatever reason and then that work will start running.*(Registered Nurse E)

Less experienced home care workers were more forthcoming in their experiences with stress and BPSD. Negative emotional reactions were reported by care workers irrespective of their clinical education, training, or professional status. By contrast, care workers with an extended working history in the sector appeared to have developed strategies for coping with this hazard:
*I used to take things a bit personally. When they’d hit me, I’d get angry. But these caregivers, they were just so gentle and loving and humorous. They would humour them and keep them in a good mood. I just had so much admiration for them. They would get beaten up on a daily basis, too.*(Registered Nurse C)

## 4. Discussion

This study sought to explore the key challenges faced by home care workers and nurses who worked with people diagnosed with dementia in their homes. The demands of their work were identified and the need to manage risks associated with occupational violence. All participants recounted incidents in their work which may be categorised as challenging or threatening, but few considered these to be occupational violence. This may be related to the subjective nature of defining occupational violence. Both registered nurses and home care workers were careful to distinguish conduct which was identified as violent from conduct which was considered hostile, aggressive or threatening, including physical and non-physical (including sexual) conduct which may be characterised as violent. Most incidents were related to interactions with the individual with dementia but some were related to interactions with the client’s family or other paid carers in the home.

All participants reported some level of exposure to BPSD and recalled verbal, physical and emotional displays which might exacerbate BPSD from homecare recipients. In particular, it was observed that forcing or compelling care recipients to participate in caregiving activities rather than negotiating or persuading them could provoke violent, aggressive or challenging reactions. Registered nurses were generally more confident in their ability to mitigate BPSD by relying on their training, clinical skills and judgment. Conversely, home care workers with at least three years of experience providing direct care and support to persons with dementia tended to rely on knowledge gained from the industry. Home care workers with minimum industry experience were less confident in their ability to manage BPSD.

All participants noted the importance of maintaining regular contact with care recipients. Frequency of contact was instrumental in being able to identify early warning signs of BPSD in the absence of thorough and current clinical documentation. Likewise, regular contact with care recipients could facilitate an understanding of the strategies that were most effective for de-escalating violent, threatening or challenging behaviour related to BPSD. Registered nurses generally perceived a professional responsibility to investigate and develop strategies to manage BPSD in the community. Home care workers on the other hand, tended to rely on a system of trial and error. Overall, home care was described as challenging and stressful, yet personally rewarding.

Whilst the focus of this paper was the occupational violence experienced by community health workers whilst caring for people with dementia, data indicated a need for training to assist them to positively manage and de-escalate challenging behaviours [19]. A person-centred care approach is advocated for the care of people with dementia, where challenging behaviours are conceptualised as expressions of unmet need. This may include neglect of psychological, social and cultural needs. For instance, one approach for minimising such behaviours is the Progressively Lowered Stress Threshold. This approach is based on the understanding that a person with dementia is increasingly less able to manage stress as dementia progresses. Therefore, people with dementia can be supported by facilitating the use of retained skills and abilities while reducing any environmental triggers such as changes in routines or carers, inappropriate levels of simulation and discomfort [30]. These principles could help community heath workers to ameliorate some of the challenging behaviours they encounter and underline the importance of further education and training.

Home care and community health work are a form of precarious work which has been associated with a range of negative work health and safety indicators [31]. This study revealed the following work-related attitudes and elements which may affect a care worker’s exposure to forms of occupational violence: lower perceived status of homecare work (in reference to rate of pay); limited training and support; belief that complaints or reports will not be followed up; fear that complaints or reports about a care recipient may affect their access to future home care services; responsibility for aggravating BPSD; resignation that violence exposure is a frequent aspect work; and willingness to return to work with an expectation of being exposed to occupational violence in the future.

The extent of the problem is obscured by the subjective nature of occupational violence, and the fact that violence-related incidents at work tend to be underreported across all industry sectors [32]. This trend is consistent with a pervasive expectation that violence is reasonably expected when performing frontline care work [33]. It is significant to note that participants expected that violent, challenging or threatening behaviour was a persistent job demand. However, both registered nurses and home care workers agreed that occupational violence should not be part of their job. This revealed a tension between norms and expectations of work.

Home care workers tended to report more episodes of occupational violence from care recipients than registered nurses. Verbal aggression was the form of violence most often perceived and reported, consistent with international literature [34,35]. Three registered nurses made comments that they believed home care workers were more frequently exposed to BPSD due to a combination of limited training, lack of quality induction and prolonged periods of time providing direct care and support.

Regardless of a care worker’s level of education, industry experience or occupational classification, there is a probability that violent incidents will not be reported unless they are perceived to be serious or equitably dealt with [36]. This has been demonstrated in homecare contexts [13] and may be due to excusing behaviours [37]; lack of certainty about whether violence has occurred [38]; organisations or colleagues may deter reporting [39]; self-blame or ‘error’ at the level of the caregiver [39]; job insecurity [40]; powerlessness [41]; normalisation of occupational violence within the industry; and excessive paperwork [42].

Since care workers perform work in isolated settings and are subject to high levels of emotional, mental, and physical demands, the nature and quality of support was important to explore. Overall, participants derived a sense of support from positive, reciprocal exchanges between themselves and clients. However, this relationship dynamic could be interrupted by sudden and unanticipated acts of violence or aggression. In such cases, home care workers felt that their relationship of mutual trust had been broken. It is also significant to note that home care workers generally reported low levels of support from their organisation. Registered nurses, however, felt they were well-supported by their respective organisations. The source(s) and significance of support for home care workers and registered nurses should be further investigated to gain insight into how best to support each group of workers.

Fear of future violence has been identified as a potential pathway linking occupational violence exposure to negative health outcomes [43,44]. In this study, a connection between exposure to workplace violence and consequent feelings of stress and anxiety was identified. Care workers were concerned about future violence because of ongoing exposure to a client with a known history of violence or aggression, uncertainty about whether an existing or future client would suddenly turn on them and a lack of preparation for the behaviours of new clients. However, care workers reported they were confident in being able to identify early warning signs that a client may become violent, threatening or challenging. Despite the persistent threat of occupational violence exposure and perceived lack of professional standing, care workers tended to report high levels of job satisfaction and an ongoing commitment to their work.

This study did not directly evaluate the relative effects of working hours and work organisation on rates of exposure to occupational violence. Some participants volunteered information about changes in the mood and behaviour of care recipients in the evening. The literature is inconclusive about whether care workers are exposed to higher rates of BPSD in the evening [6,45]. Limited evidence suggests that work organisation may affect a care worker’s exposure to BPSD, for instance caregivers who worked day shifts were more frequently exposed to violence than those working night shifts [46]. Other job design factors such as time pressure, low autonomy and insufficient staff levels have also been associated with exposure to occupational violence more generally [47,48]. Providing assistance with activities of daily living (ADL), such as bathing and showering, dressing, toileting and so on may elicit verbal and physical aggression from persons diagnosed with dementia, and as home care workers are routinely exposed to this heightened risk of violence from care recipients simply by undertaking prescribed work duties.

## 5. Study limitations

Whilst the study was successful in revealing insights into occupational violence as a job demand, the findings were limited by recall bias and voluntary participation which may have determined who was willing to disclose difficulties at work. Therefore, the results cannot be generalised to all home care and registered nurses undertaking community work with people with dementia.

## 6. Conclusions

As far as can be ascertained, these exploratory interviews represent the first qualitative study conducted in Australia which focuses on the problem of occupational violence in the health and community care sector. This research will shed light on the perspectives of health care workers and the experiences of exposure to occupational violence in the workplace that has previously gone unrecognised. The results can inform what aspects of work design can be improved to moderate the effects of occupational violence exposure or mitigate overall rates of exposure. It is imperative that community healthcare workers are provided with a safe workplace. The study findings can also be used to design training for community health care workers to assist them in anticipating and managing challenging behaviour. Further quantitative research work is needed to measure the risk factors associated with the client and the environment that contribute to health care worker’s susceptibility to exposure to occupational violence and its aftermath (such as fatigue, level of experience, training, personality, confidence, levels of burnout and access to support and clinical training). It is important that domestic legislation surrounding aged care reform takes into account the rights and interests of care workers by providing a safe system of work.

## Figures and Tables

**Table 1 ijerph-20-00438-t001:** Interview schedule.

1. In what ways have you experienced any violent or threatening behaviour from clients in your work in your clients homes?
Probes (if required): i. Would you please describe the behaviour? ii. How often does it happen? iii. Do you see it as a ‘normal’ part of your job?
2. How do you respond if a client is violent or threatening? Probes (if required):
i. Does your agency provide guidelines for managing this behaviour? ii. Do you follow the guidelines? iii. What would make it easier for you to manage violent or threatening behaviour? iv. Is this assistance provided?
3. How confident are you in identifying early signs that a client may become violent or threatening? Probes (if required):
i. Do you receive adequate training to do so? ii. What would make it easier for you to recognise early warning signs?
4. How do you feel the violence you experience affects your health and wellbeing?
5. Is there anything else you would like to add?

## Data Availability

The data presented in this study are available on request from the corresponding author. The data are not publicly available due to ethical requirements.

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
