# Peer review of "Occupational Violence Experienced by Care Workers in the Australian Home Care Sector When Assisting People with Dementia"

_ijerph, 2022, doi:10.3390/ijerph20010438_

Round 1

Reviewer 1 Report

I have carefully reviewed the manuscript title: Occupational violence experienced by care workers in the Australian home care sector when assisting people with dementia”. In its current form, the manuscript is still in a very initial stage, pretty far from the quality standards of an academic paper. So, I reject this manuscript. 

1.     In the abstract I have not seen the integration of the aims of the study, research methods, results and conclusion. I suggest the authors integrate all part and write like a story. 

2.     It is also suggested to provide a more rational background of the gap of the study and research questions. As well as explain your research questions in detail, what research will you do? I also suggest to the authors in the last paragraph of the introduction, explain the structure of the paper.

3.     The theoretical model of this study is missing. However, it is difficult to understand the concept of the paper. This study is relevant to social sciences, specifically related to psychology. So, the studies related to psychology should explain the conceptual model or theoretical model of this study. I suggest you to draw you model and explain clearly the concept of your study.

4.     The research methods is not clear and not understandable. So, I suggest follow below-mentioned suggestion. These suggestions will help to improve your study.  

l  Under the main heading of material and methods, prove the sub-heading with the title of the research approach and explain which research approach did the authors use in this study? And what strategy follow to write the research methods?

l  The target population of this study is not clear. The authors have explained the interviews are conducted among the community care agencies across two states (South Australia and New South Wales). Describe in detail why you select the community care agencies to conduct research, what is the rational background explain in detail.

l  It is also suggesting the author provide in detail what is the validity and reliability of the questions that the authors ask from the target population. It is also suggesting the authors prove the pre-test details of the questions that you ask in the interviews.

l  The measurement of the variables is also not well explained. I want to see the logical reasons how items of the interview questionnaire were valid and what was the sample items of the variables. Explain each variable measurement.

5.     Findings/Results. I suggest you follow the below-mentioned comments and improve your results.  

l  The description of the statistical analysis is not given. There is no description of how analysis the data.  I suggest you for analysis use any authentic soft wear i.e. NVivo, ATLAS. Ti or Provalis Research Text Analytics Software.     

6.     The conclusion part of this study is also feeble. I suggest you re-write your conclusion and explain the novel contribution. Furthermore, the conclusion must be integrated with your study's introduction, theory, results and findings.

Author Response

  1. Aims of the study stated in abstract: "This study aimed to explore the challenges faced by home care workers and nurses working with people diagnosed with dementia in the community, to identify job demands contributing to their vulnerability to occupational violence, and to determine ways to help manage occupational violence. " Method stated in abstract: "A qualitative descriptive study was conducted by interviewing 10 homecare workers and six registered nurses from agencies in South Australia and New South Wales, Australia." Results in abstract: "The following themes were identified: i) sources of threats; iii) categories of violent, threatening or challenging behaviour; iv) aggravating factors; v) early warning signs; vi) education and training." Conclusion in abstract: "Serious issues were identified by participants, yet very little is known about occupational violence for these community care workers. Findings can inform what aspects of work design can be improved to moderate the effects of occupational violence exposure or mitigate rates of exposure, to enable long-term services for people with dementia." Reviewer suggests this should be written as a story - not sure how as this follows standard academic practice.
  2. This author requests more background whereas another reviewer suggests less. 

Reviewer 2 Report

This is an interesting article presenting an exploratory study in which the authors research the challenges faced by home care workers and nurses working with people diagnosed with dementia in order to identify job demands contributing to their vulnerability to occupational violence, and to determine ways to help manage occupational violence. This topic is of great interest and the implications from the research described in the article go beyond the national borders of the country in which it was conducted, even though the sample used is small (10 homecare workers and six registered nurses).

The article is very clear and well structured in its methodological choices and arguments. The authors poses some very interesting findings and implications for care workers and their organisations. I have only two remarks:

- line 375 "Consequently, approximately 50% admitted to feeling": "half" should replace "50%" because the sample is very small

- p. 17 Conclusion: a few further thoughts on how this research could impact on practitioners and organisations would enrich the article.

Author Response

Thank you for your positive review of the article. 

Half replaces 50% as advised.

The conclusion has been adapted. 

Reviewer 3 Report

The article is clear and the study in very interesting. I I would like to congratulate the authors for their work and hope that they can continue their research, maybe expanding the sample. 

I have only one suggestion for visual presentation: the direct speech of nurses and health care worker maybe be inserted in a summary table, to make the article more flowing. It is just a proposal. 

Kind regards.

Author Response

Thank you for the positive review.

I believe that a table containing the quotes would be repetitive.

Reviewer 4 Report

The authors presented a study that explores the challenges faced by home care workers and nurses working with people diagnosed with dementia in the community, identifies job demands contributing to their vulnerability to occupational violence, and determines ways to help manage occupational violence. It is a very interesting study, however, the authors can, in general, improve the manuscript, reducing significantly the introduction to essentials, and reorganizing the presentation of the results.

Author Response

Thank you for the review.

Unfortunately, there were no specific examples given of where the manuscript could be improved.  

This reviewer suggests shortening the introduction, whereas others request more information. 

The presentation of results follow usual academic presentation of qualitative research, so it is not clear how this could be improved.

Reviewer 5 Report

This is incredibly important work. Thank you for tackling such a difficult topic. Healthcare workers are subjected to physical violence at alarming rates and it is rarely discussed. Drawing attention to this issue may result in positive change.

Author Response

Thank you for the positive review.

No specific improvements are suggested.

Round 2

Reviewer 1 Report

Thank you very much to prove the opportunity to review the manuscript title: Occupational violence experienced by care workers in the Australian home care sector when assisting people with dementia”. In its current form, the manuscript is still in a very initial stage, pretty far from the quality standards of an academic paper. So, I reject this paper.